# Preventive Effects of Anthocyanins from *Lycium*
*ruthenicum* Murray in High-Fat Diet-Induced Obese Mice Are Related to the Regulation of Intestinal Microbiota and Inhibition of Pancreatic Lipase Activity

**DOI:** 10.3390/molecules27072141

**Published:** 2022-03-26

**Authors:** Na Li, Xi Liu, Jing Zhang, Yan-Zhi Lang, Lu Lu, Jia Mi, You-Long Cao, Ya-Mei Yan, Lin-Wu Ran

**Affiliations:** 1Key Laboratory of Environmental Factors and Chronic Disease Control, School of Public Health and Management, Ningxia Medical University, Yinchuan 750004, China; 20200310247@nxmu.edu.cn (N.L.); liuxi950321@163.com (X.L.); asbdx242322@163.com (J.Z.); langyanzhi2020@163.com (Y.-Z.L.); 2Goji Berry Research Institute, Ningxia Academy of Agriculture and Forestry Sciences, Yinchuan 750002, China; lulubing8901@163.com (L.L.); lorna0102@126.com (J.M.); youlongchk@163.com (Y.-L.C.); 3Laboratory Animal Center, Ningxia Medical University, Yinchuan 750004, China

**Keywords:** *L. ruthenicum*, obesity, intestinal microbiota, pancreatic lipase

## Abstract

*Lycium**ruthenicum* Murray (*L*. *ruthenicum*) has been used both as traditional Chinese medicine and food. Recent studies indicated that anthocyanins are the most abundant bioactive compounds in the *L*. *ruthenicum* fruits. The purpose of this study was to investigate the preventive effects and the mechanism of the anthocycanins from the fruit of *L*. *ruthenicum* (ACN) in high-fat diet-induced obese mice. In total, 24 male C57BL/6J mice were divided into three groups: control group (fed a normal diet), high-fat diet group (fed a high-fat diet, HFD), and HFD +ACN group (fed a high-fat diet and drinking distilled water that contained 0.8% crude extract of ACN). The results showed that ACN could significantly reduce the body weight, inhibit lipid accumulation in liver and white adipose tissue, and lower the serum total cholesterol and low-density lipoprotein cholesterol levels compared to that of mice fed a high-fat diet. 16S rRNA gene sequencing of bacterial DNA demonstrated that ACN prevent obesity by enhancing the diversity of cecal bacterial communities, lowering the *Firmicutes*-to-*Bacteroidota* ratio, increasing the genera *Akkermansia*, and decreasing the genera *Faecalibaculum*. We also studied the inhibitory effect of ACN on pancreatic lipase. The results showed that ACN has a high affinity for pancreatic lipase and inhibits the activity of pancreatic lipase, with IC50 values of 1.80 (main compound anthocyanin) and 3.03 mg/mL (crude extract), in a competitive way. Furthermore, fluorescence spectroscopy studies showed that ACN can quench the intrinsic fluorescence of pancreatic lipase via a static mechanism. Taken together, these findings suggest that the anthocyanins from *L. ruthenicum* fruits could have preventive effects in high-fat-diet induced obese mice by regulating the intestinal microbiota and inhibiting the pancreatic lipase activity.

## 1. Introduction

Obesity is characterized as a serious global epidemic by WHO. In recent years, the prevalence of overweight/obesity in China has increased year by year. China has the largest number of affected people worldwide, where approximately 46% of adults and 15% of children are obese or overweight [1]. Obesity is generally considered a lifestyle-associated medical condition, with an increased risk of chronic diseases, such as type 2 diabetes, cardiovascular diseases, non-alcoholic fatty liver, and complex metabolic and intestinal flora disorders [2,3]. Therefore, researchers have been looking for convenient, safe, and effective methods to prevent and treat obesity.

Pancreatic lipase is secreted by the pancreas of mammals and then released into the gastrointestinal system. It works synergistically with the bile salts secreted by the liver to break down fat into fatty acids and glycerol [4]. Triglyceride, with high calorie characteristics, is the major energy source for humans. Therefore, the inhibition of pancreatic lipase activity is considered to be one of the most important therapies for preventing obesity [5]. Orlistat, a commonly used drug for obesity treatment, can selectively inhibit pancreatic lipase and prevent 30% of dietary fat absorption [6]. Nevertheless, Orlistat therapy may be accompanied by multiple adverse reactions. Recently, various inhibitors of pancreatic lipase activity have been explored and natural compounds have attracted worldwide attention due to their excellent inhibitory effects and low toxic effects [7].Plant bioactive compounds have been extensively used to promote human health. Among these, anthocyanin-rich foods have appeared as promising therapeutics for obesity [8].

Intestinal microbiota, also referred to as a hidden organ, contain tens of trillions of microbes in the human intestine, which are associated with obesity [9].The diet and surrounding environment are closely related to gut microbes. Imbalances in the structure of the gut microbiota induced by high-fat diet consumption may impair gut barrier function, which provokes obesity and metabolic disease. Recent studies have shown that natural products, such as flavonoids, have a protective effect on obesity [10].

Anthocyanin, a special kind of flavonoid widely found in the epidermal tissues of plants, is innocuous and responsible for the distinct colors, such as red, purple, or blue, of many flowers, fruits, and vegetables [11]. Various anthocyanins from different sources have been reported to affect the viability of colonic bacterial groups, implying that dietary modulation with anthocyanins may play a role in reshaping the gut microbial community to provide beneficial effects, such as weight loss [12].

*Lycium**ruthenicum* Murray (*L. ruthenicum*), a member of *Lycium* genus in the family *Solanaceae*, is a long-living perennial shrub native to northwestern China, and is a newly discovered resource of *Lycium*
*barbarum* L. with high economic value [13]. It has been reported that the fruits of *L. ruthenicum* are rich in anthocyanins. 3-*O*-[6-*O*-(4-*O*-(trans-p-coumaroyl)-a-l-rhamnopyranosyl)-b-d-glucopyranoside]-5-*O*- [b-d-glucopyranoside] (Appendix A) is considered to be the main component of anthocyanin from the fruits of *L. ruthenicum* [13,14]. It has been reported that anthocyanins from the fruits of *L. ruthenicum* (ACN) possess a wide range of biological functions, including antioxidant and anti-inflammatory properties and can change the gut microbiota and reduce the risk of chronic diseases, such as obesity [15,16,17]. However, the role of ACN in the prevention of obesity is still not fully understood. In this study, we evaluated the preventive effects of ACN on obesity induced by a high-fat diet. The effect of drinking ACN on intestinal microbiota in high-fat-induced obese mice was explored. Furthermore, the pancreatic lipase inhibitory activities of ACN and its main compound were also measured to provide the possible mechanism.

## 2. Results

### 2.1. Anthocyanins Extract from the Fruits of L. ruthenicum Reduced the Body Weight of High-Fat-Fed Mice

The food intake and water consumption of the mice were measured throughout the whole experiment. During the 14-week treatment period, no significant differences were observed in the food intake (Figure 1a) and water consumption (Figure 1b) between the HFD group and HFD + ACN group. As depicted in Figure 1c, high-fat-fed mice gained more body weight than control mice. Nevertheless, after 14 weeks, high-fat-diet-fed mice were treated with 0.8% ACN in their drinking water gained bodyweight more slowly than the high-fat group.

### 2.2. Effects of Anthocyanin Extract from the Fruits of L. ruthenicum on the Serum Lipid Levels in High-Fat-Diet-Fed Mice

The changes in the serum lipid levels of mice after 14 weeks of being fed the high-fat diet are shown in Figure 2. The levels of TC and LDL-C were significantly reduced in the HFD + ACN group when compared to the HFD group. However, the levels of HDL-C in the three groups were not statistically significant different. These results indicate that the anthocyanin extract from the fruits of *L. ruthenicum* can alleviate high-fat-diet-induced dyslipidemia in mice.

### 2.3. Anthocyanin Extract from the Fruits of L. ruthenicum Ameliorated High-Fat-Diet-Induced Lipid Accumulation in the Liver and Adipose Tissues in Mice

The effects of ACN on lipid accumulation in the liver and adipose tissues were determined by H&E and Oil red O staining. In comparison with the control group, the mice fed the high-fat diet showed significantly increased lipid accumulation in the liver and enlarged adipocyte cells in the adipose tissue. When compared to the HFD group, body fat accumulation in high-fat-fed mice treated with ACN was reduced, especially in the liver and epididymal adipose tissue (Figure 3). These results are consistent with the results of the serum biochemical indicators, indicating that ACN from the fruits of *L. ruthenicum* could prevent and/or treat high-fat-diet-induced hyperlipidemia and obesity in mice.

### 2.4. Anthocyanins Extract from the Fruits of L. ruthenicum Regulated the Intestinal Microbiota in High-Fat-Fed Mice

To assess the effect of ACN on the intestinal microflora in high-fat-diet-fed mice, cecal samples were collected and sequenced for the V4 variable regions of 16S rDNA gene. After data trimming and quality filtering, high-throughput sequencing yielded about 63,610, 63,364, and 61,244 effective tags in 3 groups, respectively. The results reveal that more than 60% of the original data were considered valid (Table 1). The richness and diversity of the cecal microbes in the three groups are assessed by the α-diversity index in Table 2. Among them, the Shannon index and Chao1 index in the HFD group were significantly lower than that in the control group and these indexes in the ACN treatment group were slightly improved compared to the HFD group, but the difference did not reach significance. The principal coordinate analysis (PCoA) plot based on the distance of the unweighted UniFrac (Figure 4a) showed greater similarity between bacterial communities from the ACN treatment group and control group than between communities from the HFD group and control group.

In terms of the bacterial composition at the phylum level, *Firmicutes* was the most dominant bacterial phyla in the cecal samples of mice, followed by *Bacteroidota*, *Verrucomicrobiota*, *Desulfobacterota*, unidentified_Bacteria, *Actinobacteriota*, *Campylobacterota*, and *Deferribacteres* (Figure 4b). Among them, the relative abundance of *Firmicutes* was significantly increased in the HFD group mice when compared to the control group (Figure 4c). In contrast, ACN from the fruits of the *L. ruthenicum* treatment correctively decreased the level of *Firmicutes* and increased the level of *Bacteroidota* when compared to the HFD group(Figure 4d). As a result, the ratio of *Firmicutes*/*Bacteroidota* in the ACN treatment group was significantly lower than that of the HFD group (Figure 4e). Moreover, the relative abundance of *Verrucomicrobiota* in the ACN treatment group was significantly increased compared with the HFD group (Figure 4f). At the genus level, the ACN treatment led to decreases in both the *Faecalibaculum* and *Blautia* abundances and increases in the *Akkermansia* abundances compared with the HFD group (Figure 5). Therefore, the bodyweight-lowering effect of ACN from the fruits of *L. ruthenicum* was related to the reduction in the *Firmicutes*/*Bacteroidota* ratio, the increase in the genera *Akkermansia*, and the decrease in the genera *Faecalibaculum*.

### 2.5. Effects of Anthocyanins from the Fruits of L. ruthenicum on Pancreatic Lipase Activity

The effects of anthocyanins on the hydrolysis activity of pancreatic lipase were investigated and the results are presented in Figure 6a. It shows that both the main compound and crude extract of anthocyanins inhibited the activity of lipase in a dose-dependent manner. The IC_50_ values of the main compound and crude extract of anthocyanins were 1.80 ± 0.20 and 3.03 ± 0.43 mg/mL, respectively (Table 3). This means the main compound anthocyanin from the fruits of *L. ruthenicum* might be a major contributor to the lipase inhibition.

The inhibitory type of anthocyanins from the fruits of *L. ruthenicum* on pancreatic lipase is competitive and reversible inhibition (Figure 6b). The inhibitory effect of the anthocyanins against pancreatic lipase was determined using different concentrations (5, 10, 15, and 20 mg/mL) of pancreatic lipase solution. The *Lineweaver–Burk* curve of inhibition kinetics of pancreatic lipase is shown in Figure 6c. The kinetic parameters of the *K_m_* and *V*_max_ values for the samples are listed in Table 3. In the presence of the main compound and crude extract of anthocyanins, the *K_m_* values increased while the *V*_max_ remained unchanged, indicating their enzymatic inhibitory type was competitive [18].

### 2.6. Effects of Anthocyanins from the Fruits of L. ruthenicum on the Fluorescence Spectra of Pancreatic Lipase

To explore the inhibiting mechanism of anthocyanins from the fruit of *L. ruthenicumon* pancreatic lipase, the effects of anthocyanins on the intrinsic fluorescence spectra of pancreatic lipase are shown in Figure 7. The fluorescence intensity of pancreatic lipase decreased with the increase in the concentration of crude anthocyanins. The same pattern of results was observed in the main compound anthocyanin treatment group. In addition, slight red shifts in the maximum fluorescence of pancreatic lipase were recorded both in the presence of crude anthocyanin extract and the main compound anthocyanin (Figure 7a,b). These results suggest that the anthocyanins from the fruit of *L. ruthenicum* could interact with tryptophan (Trp) residues of pancreatic lipase and quench the fluorescence of lipase.

To predict the possible quenching mechanism, the effects of anthocyanins on the lipase were described by the *Stern–Volmer* curve (Figure 7c,d). The values of *K_sv_* and *K_q_*, which were determined with the *Stern–Volmer* curve by linear regression of plots of *F*_0_/*F* vs. [*Q*] at different temperatures, are shown in Table 4. The values of *K_sv_* decreased accompanied by an increase in the temperature, which showed that the type of fluorescence quenching of the pancreatic lipase by anthocyanins was static quenching.

The double logarithmic curve of the plots of Lg [(*F*_0_ − *F*)/*F*] vs. Lg [*Q*] (Figure 7e) was used to calculate the values of *K_a_* and *n* (Table 4). Obviously, the inhibitory activity of the main compound anthocyanin on the pancreatic lipase was higher than the crude extract of anthocyanins. In other words, the main compound anthocyanin has strong binding sites on pancreatic lipase.

To determine the interaction between anthocyanins and pancreatic lipase, the thermodynamic parameters of Δ*G*, ΔS, and Δ*H* are shown in Table 5. The thermodynamic parameters of the interaction between anthocyanins and pancreatic lipase were Δ*G* < 0, Δ*S* < 0, and Δ*H* < 0, which indicates that the major intermolecular forces between anthocyanins and pancreatic lipase were hydrogen bonds and Van Der Waals force.

## 3. Discussion

Anthocyanin is an important plant pigment, which has been associated with a reduced risk of developing chronic diseases [19,20]. In this study, we focused on the effects of anthocyanins from the fruit of *L. ruthenicum* on the development of obesity induced by a high-fat diet.

The present investigation confirmed that a high-fat diet can induce a significant increase in body weight. After treatment with anthocyanins from the fruits of *L. ruthenicum*, weight loss in the high-fat-diet-fed mice was observed, although there was no significant difference in the food intake and water consumption between the HFD group and the anthocyanin extract treatment group. The results correspond well with previous studies that found that the consumption of anthocyanin-rich purple corn could effectively reduce weight growth in high-fat-diet-fed mice [21]. Furthermore, liver histopathological analysis showed that the liver steatosis was improved after treatment with anthocyanins from the fruits of *L. ruthenicum*. As reported, the anthocyanins from purified mulberry could effectively restrain lipid accumulation in the liver [22]. Similarly, histopathological analysis showed that high-fat-diet-induced obese mice by showed a smaller-sized adipocyte cells after treatment with anthocyanins from *L. ruthenicum*. Wu et al. reported that supplementation with an anthocyanin-rich extract improved the lipid profile by decreasing the serum triglyceride and total cholesterol levels in mice [8]. It was also found in this study that the anthocyanin extract from *L. ruthenicum* effectively decreased serum cholesterol levels in high-fat-diet-fed mice. Noticeably, all of the effects exerted by anthocyanins from the fruits of *L. ruthenicum* did not cause abnormal clinical signs throughout the 14-week experiment. These results suggest that anthocyanins from the fruits of *L. ruthenicum* have beneficial effects on lipid profiles and can counteract dyslipidemia induced by a high-fat diet.

Microorganisms in the large intestine play an important physiological role in vital processes, such as digestion, vitamin synthesis, and metabolism [23]. The complex interaction between diet and gut microbiota may contribute to an individual’s overall health and the incidence of chronic disorders, such as obesity. It has been reported that only a small part of anthocyanins from *L. ruthenicum* were absorbed in the upper gastrointestinal while a large part of anthocyanins entered the large intestine and were metabolized there by intestinal microbiota, which promotes the production of short-chain fatty acids to maintain intestinal health [24]. At the phylum level, *Firmicutes* and *Bacteroidota* were the most abundant in host metabolism, which involves anthocyanin uptake and degradation. In addition, the ratio of *Firmicutes* to *Bacteroidota* is considered to be closely related to obesity. Studies have shown that obese individuals have a higher ratio of *Firmicutes*/*Bacteroidota* than normal weight controls [25]. Zhang et al. reported that the relatively high level of *Firmicutes* and low level of *Bacteroidota* may lead to an increase in the ability of obese microbiota to obtain energy from the diet [26]. In this study, we investigated the preventive effects of anthocyanins from *L. ruthenicum* on high-fat-diet-associated gut microbiota disturbances. Interestingly, we found that there were significant changes in the intestinal microbiota composition in the three groups. The anthocyanin extract from the fruits of *L. ruthenicum* decreased the abundance of *Firmicutes* and the ratio of *Firmicutes*/*Bacteroidota* in cecal samples compared with the high-fat-diet group. At the genus level, the bacterium of *Faecalibaculum* belongs to *Firmicutes*. Studies have shown that *Faecalibaculum* is related to obesity induced by a high-fat diet [27]. The anthocyanin extract from the fruits of *L. ruthenicum* led to a decrease in the abundance of *Faecalibaculum* compared with the high-fat-diet group. In addition, *Verrucomicrobiota* is also an important bacterium at the phylum level, preceded only by *Bacteroidota* and *Firmicutes*. In this study, we found that the level of *Verrucomicrobiota* in the anthocyanins-extract-treated mice was higher than that in the high-fat-diet-fed mice, which may be related to the increase in the relative abundance of *Akkermansia* at the genus level. *Akkermansia*
*muciniphila* is a colonic mucin-degrading bacterium believed to have beneficial effects on gastrointestinal health, particularly in the context of obesity [28]. Therefore, the bodyweight-lowering effect of the anthocyanin extract from the fruits of *L.*
*ruthenicum* was related to the reduction in the *Firmicutes*/*Bacteroidota* ratio, the increase in the genera *Akkermansia*, and the decrease in the genera *Faecalibaculum*.

Pancreatic lipase is a key enzyme involved in hydrolyzing triacylglycerol into monoacylglycerols and non-esterified fatty acids, which may form micells that serve as necessary intermediates for cholesterol uptake in enterocytes. Our results indicate that the anthocyanins of the crude extract and the main compound from the fruits of *L. ruthenicum* have competitive inhibitory activity against pancreatic lipase, with IC50 values of 3.03 and 1.80 mg/mL, respectively. The relatively low IC50 value indicates that anthocyanins from the fruits of *L. ruthenicum* can be used as a preventive agent to inhibit high-fat-diet-induced obesity. Regarding the mechanism, anthocyanins from the fruits of *L. ruthenicum* could interact with the pancreatic lipase amino acid residue Trp. Our results are in accordance with those reported by Zeng et al., who found that the amino acid residue Trp is the main interaction site found between polyphenolic compounds and pancreatic lipase [29]. Furthermore, the quenching rate constant *K_q_* was used to identify whether the quenching was dynamic or static [30]. Here, the *K_q_* values (Table 4) of anthocyanin from *L. ruthenicum* were higher than the maximum value (2.0 × 10^10^ L/(mol·s)) of the quenching rate constant in dynamic quenching [31], which indicates that the quenching type is an intrinsic and static process. Static quenching has previously been reported for the interaction of pancreatic lipase with polyphenols [32]. In addition, the type of binding force between anthocyanins and pancreatic lipase was evaluated using thermodynamics values. The negative Δ*H* indicates the exothermic characteristic of the interaction, the negative Δ*S* suggests a reduction in the randomness at the interface, and both a negative Δ*H* and Δ*S* value indicates that van der Waals force and hydrogen bonds are the main acting force [33,34]. The negative Δ*S* and negative Δ*H* (Table 5) indicate that the interaction between anthocyanins from *L. ruthenicum* and pancreatic lipase is spontaneous and exothermic, accompanied by hydrogen bonds and Van Der Waals force binding. Our experiment did not evaluate the effect of anthocyanins from *L. ruthenicum* on the activity of pancreatic lipase in vivo, and further studies about the anthocyanins against pancreatic lipase are required.

## 4. Materials and Methods

### 4.1. Materials

Fresh fruits of *L. ruthenicum* were collected from the resources of the National Engineering Technology Research Center (Yinchuan, China) and stored at −20 °C until use. A high-fat diet MD12015 was purchased from Medison Company (Yangzhou, China). 4-nitrophenyl laurate was purchased from Tokyo Industrial Company (Tokyo, Japan). Porcine pancreatic lipase was acquired from Sigma-Aldrich Co., Ltd. (St. Louis, MO, USA). Crystalline sodium acetate was obtained from Shanghai Guangnuo Chemical Technology Co., Ltd. (Shanghai, China). Formic acid, trifluoroacetic acid (TFA), and dichloromethane (chromatographic purity) were purchased from Mreda Technology Co., Ltd. (Beijing, China). High-density lipoprotein (HDL-C), low-density lipoprotein (LDL-C), total cholesterol (TC), and triglyceride (TG) kits were acquired from Nanjing Jiancheng Biological Institute (Nanjing, China). Oil red O powder was purchased from Sigma-Aldrich Co., Ltd. (St. Louis, MO, USA). Hematoxylin and eosin (H&E) staining solution was obtained from Solarbio Science & Technology Co., Ltd. (Beijing, China). Other chemicals and agents were of analytical purity.

### 4.2. Preparation of the Anthocyanins from L. ruthenicum

Total anthocyanins were extracted from *L. ruthenicum* according to the method reported by Yan et al. [35]. In brief, 0.5% TFA methanol solution was used to extract anthocyanins from *L. ruthenicum*. After vacuum suction filtration, the extracted solution was concentrated under reduced pressure at a temperature lower than 40 °C, and the residue was freeze-dried to prepare the crude extract of anthocyanins (ACN). Further purification and separation of anthocyanins were performed using an XAD-7 macroporous resin column and sephadex LH-20, respectively. The crude extract solution was passed through an XAD-7 macroporous resin column and eluted by the addition of 0.5% (*v*/*v*) TFA in methanol. Then, the anthocyanins from XAD-7 were passed through a sephadex LH-20 gel column and eluted by the addition of 0.5% (*v*/*v*) TFA in 70% methanol (*v*:*v*). The recovery from sephadex LH-20 was passed through AKTA semi-preparative high-performance liquid chromatography (Agilent Technologies, Santa Clara, CA, USA) for the preparation of the main compound of anthocyanin. The mobile phase A was methanolic acetonitrile solution (*v*:*v* = 1:1) and the mobile phase B was 5% aqueous formic acid. Elution with 40% phase B occurred at a flow rate of 1.2 mL/min. The main compound was identified by matrix-assisted laser desorption ionization time-of-flight mass spectrometry (MALDI-TOF-MS). The main anthocyanin compound from the fruits of *L. ruthenicum* was 3-*O*-[6-*O*-(4-*O*-(trans-p-coumaroyl)-a-l-rhamnopyranosyl)-b-d-glucopyranoside]-5-*O*-[b-d-glucopyranoside] (Appendix A), which is consistent with previous results [13,14].

### 4.3. Animals and Samples

A total of 24 male C57BL/6J mice (aged 6 weeks) were purchased from Jiangsu Jicui Yaokang Biotechnology Co., Ltd. (Nanjing, China) and maintained in a room with an alternating 12 h day/night cycle at 22 ± 2 °C and free access to food or water. After a 5-day acclimatization period, mice were randomly divided into 3 groups (8 mice per group). The control group of animals were given a normal diet (containing 28.1 kcal% fat, 15.6 kcal% carbohydrate, and 56.3 kcal% protein) while the high-fat-diet group (HFD) and anthocyanins group (HFD + ACN) were fed a high-fat diet of MD12015 (containing 40.3 kcal% fat, 42.8 kcal% carbohydrate, and 16.9 kcal% protein, as shown in Table 6) for 14 weeks. The mice had free access food and water. Among them, the HFD + ACN group of animals were allowed to drink pure water containing 0.8% crude extract of anthocyanins from the fruits of *L. ruthenicum*. During the experiment protocol, the bodyweight, food intake, and water consumption of the mice were recorded once a week. At the end of the experiment, the mice were fasted for 10 h and sacrificed with a carbon dioxide animal asphyxiator. Blood samples were collected from the heart using a 1 mL syringe. After standing at room temperature for 2 h, the whole blood samples were centrifuged (4000× *g*, 10 min, 4 °C) to obtain serum, and stored at −80 °C for further biochemical analysis. Fresh cecal digesta samples were collected and then the samples were stored at −80 °C immediately until the intestinal microbiota analysis. The liver samples were removed immediately and divided into two parts. Among them, one part was washed with normal saline and placed in 4% paraformaldehyde for 24 h before H&E staining; the other part was frozen at −80 °C with OTC compound for Oil red O staining. Similarly, the adipose tissues were weighed and washed with normal saline for H&E staining. All animal experiments were approved by Ningxia Medical University Animal Ethics and Welfare Committee (NCDWZX-2018-096).

### 4.4. Determination of Serum Lipid Levels

The serum concentrations of TC, TG, HDL-C, and LDL-C were measured using the procedure provided by the supplier of the commercial assay kits.

### 4.5. Calculation of the Adipose Coefficient

At the end of the experiment, the mesenteric, epididymal, and perirenal adipose tissues were collected immediately. Then, the adipose coefficient was calculated the formula: adipose weight (mg)/body weight (g).

### 4.6. Histological Observation of Liver and Adipose Tissue

After fixation in 4% paraformaldehyde, the liver and adipose samples were extensively washed in running tap water, dehydrated in an automatic tissue dehydrator ASP300, and embedded in paraffin. Then, 5-μm sections from the paraffin-embedded tissue were stained with H&E. Briefly, sections were deparaffinized using xylene and washed with graded ethanol. Then, the sections were stained with hematoxylin solution for 4 min, washed in running tap water for 10 min, and counterstained with eosin solution for 2 min. Finally, sections were washed in pure water for 5 min, dehydrated in a series of ethanol/water, and cleared in xylene. The sections were observed by optical microscopy.

Oil red O staining was performed as described by Nam et al. [36]. Briefly, frozen sections were made using a Leica Cryostat at a thickness of 5 μm, fixed with 4% paraformaldehyde for 30 min, and washed in running tap water for 5 min. Then, sections were stained with freshly prepared Oil Red O solution for 10 min, washed in 60% isopropanol for few seconds, and dipped for 3 min in hematoxylin. Finally, representative images were acquired using optical microscopy from Nikon (Tokyo, Japan).

### 4.7. Bacterial DNA Extraction, 16S rRNA Gene Sequencing, and Biological Information Analysis

Total genome DNA was extracted from the cecal samples according to the cetyl trimethyl ammonium bromide/sodium dodecyl sulfate (CTAB/SDS) method reported by Bansal et al. [37]. The DNA concentration and purity were monitored on 1% agarose gels and diluted to 1 ng/μL as the template. The bacterial 16S rDNA gene was amplified using the specific primer with a barcode via PCR: 341F 5′-CCTAYGGGRBGCASCAG-3′ and 806R 5′-GGACTACNNGGGTATCTAAT-3′. Briefly, thermal cycling consisted of initial denaturation at 95 °C for 3 min, followed by 25 cycles of denaturation at 95 °C for 30 s, annealing at 55 °C for 30 s, and elongation at 72 °C for 45 s. Then, final extension was performed at 72 °C for 20 min. After purification, quantification, and quality inspection, PCR products were sequenced on an Illumina Miseq platform with the paired-end method at Novogene Co., Ltd. (Beijing, China).

The raw sequencing data of 16S rDNA sequencing were spliced, filtered, and interfering data removed by Cutadapt (V1.9.1) (https://pypi.org/project/cutadapt/, accessed on 20 February 2022) to obtain the clean reads. The effective tags from each sample were clustered into operational taxonomic units (OTUs) based on the 97% sequence identity using UPARSE software (UPARSE v7.0.1001) (www.drive5.com/uparse/, accessed on 20 February 2022) and the representative sequence of each OTU was aligned against the database for taxonomy analysis [38]. For α-diversity analysis, the community richness and community diversity indices were calculated using QIIME (Version 1.9.1) (https://qiime2.org, accessed on 20 February 2022). The β-diversity of the microbial communities was explored with principal co-ordinates analysis (PCoA) plots based on the unweighted UniFrac distance using R software (Version 2.15.3) (https://www.r-project.org, accessed on 20 February 2022).

### 4.8. Pancreatic Lipase Activity Assay

The pancreatic lipase activity was determined according to a slightly modified method reported by McDougall et al. [39]. Lipase from porcine pancreatic (type II) was dissolved in ultrapure water at 5 mg/mL, and the supernatant was used as a lipase solution after being centrifuged at 12,000× *g* for 3 min. The pancreatic lipase substrate solution was 0.08% 4-nitrophenyl laurate dissolved in 5 mmol/L sodium acetate solution (containing 1% triton X-100). The sample solutions of the crude extract of anthocyanins and the main compound anthocyanin from the fruits of *L*. *ruthenicum* were dissolved in ultrapure water at different concentrations (0, 0.4, 0.6, 0.8, 1.0, 1.5, 2.0, and 4.0 mg/mL). For the inhibition rate studies, 10 μL of sample solution, 20 μL of phosphate buffer (pH 7.2–7.4), and lipase solution were mixed and preheated at 37 °C for 10 min in a 96-well microplate. After incubation, 40 μL of substrate solution were added to initiate the colorimetric reaction. The mixture was incubated at 37 °C for 2 h and cooled to room temperature. Finally, the lipase activity was determined by measuring the yellow chromogen *p*-nitrophenol at 405 nm using a spectrophotometer. All samples were assayed in triplicate and a control blank was prepared for each sample. The inhibition rate was calculated according to Formula (1):Inhibition rate (%)= [1 − (*b* − *B*)/(*a* − *A*)] × 100,(1)
where *A* is the absorbance of the control blank, *a* is the absorbance without anthocyanins, *B* is the absorbance of the group without pancreatic lipase, and *b* is the absorbance of the sample.

### 4.9. IC_50_, Inhibition Type, and Ki Values

The concentrations of the crude extract of anthocyanins and the main compound from the fruits of *L*. *ruthenicum* giving 50% inhibition of lipase activity (IC_50_) were calculated by multiple linear fitting, using the probit regression equation model.

The inhibition type of the anthocyanins on the activity of pancreatic lipase was identified according to the method reported by Yoshioka et al. [40] using the reaction rate *V* as the ordinate and the pancreatic lipase concentration as the abscissa, and the inhibition type was judged according to the slope of the straight lines. The kinetic measurements were conducted using the same method described for the inhibition assays, except that the concentration of the pancreatic lipase solutions (5, 10, 15, and 20 mg/mL) were different and the reaction time was 15 min.

The inhibitory kinetics parameters of the pancreatic lipase by anthocyanins were determined using the *Lineweaver**–Burk* equation [41].The lipase activities were assayed at different 4-nitrophenyl laurate concentrations (0.4, 0.6, 0.8, 1.0, 1.2, 1.4, and 1.6 mg/mL). Then, the *Lineweaver–Burk* double reciprocal curve was constructed by plotting the reciprocal of the substrate concentration [*S*] against the reciprocal of the enzyme reaction velocity *V*. From the *Lineweaver–Burk* plots (2), the Michaelis constant (*K_m_*) and maximum velocity (*V*_max_) can be determined:(2)$$\frac{1}{V}=\frac{K_{m}}{[S]K_{i}V_{\max}}[I]+\frac{1}{V_{\max}}\left(1+\frac{K_{m}}{\left[S\right]}\right),$$
where *K_i_* is the dissociation constant and *K_m_* is the Michaelis constant. *V*_max_ is the maximum rate of the enzymatic reaction and [*I*] is the concentration of the anthocyanins solution.

### 4.10. Fluorescence Spectroscopy Measurement

The fluorescence spectroscopy was obtained using a F-4600 fluorescence spectrofluorometer from Hitachi Corporation (Tokyo, Japan) with a fixed concentration of pancreatic lipase (2.0 mL of 5 × 10^−6^ mol/L) in the presence of different volumes of anthocyanins. Briefly, the final concentration rates of the anthocyanins and pancreatic lipase were 0, 0.5:1, 1:1, 2:1, 4:1, 5:1, 10:1, and 20:1, respectively. Then, the solutions were mixed at 20 or 30 °C. The excitation wavelength was set at 280 nm and the emission spectra were scanned in the range of 300 to 400 nm. Finally, fluorescence quenching is usually described by the linear *Stern–Volmer* equation:*F*_0_*/F* = 1 + *K_q_τ*_0_ [*Q*],(3)
where *F*_0_*/F* is the intensity ratio in the absence or presence of quencher (anthocyanins), [*Q*] is the concentration of anthocyanins, *τ*_0_ is the average fluorescence lifetime (about 10^−8^ s), *K_q_* is the bimolecular quenching constant, and *K_sv_* is the *Stern-Volmer* quenching constant calculated by *K_q_τ*_0_.

For the fluorescence quenching mechanism, the apparent binding constant (*K_a_*) between anthocyanins and pancreatic lipase and the number of binding sites (*n*) were calculated from the static quenching formula:Lg [(*F*_0_ − *F*)/*F*] = Lg *K_a_* + *n*Lg [*Q*],(4)

### 4.11. Thermodynamic Parameters Assessment

The thermodynamic parameters of the interaction between the anthocyanins and pancreatic lipase, including the free energy change (Δ*G*), enthalpy (Δ*H*), and entropy (Δ*S*), were calculated according to Equations (5)–(7):Δ*G* = −R*T*ln*K*,(5)
Δ*H* = [ln(*K_a_*_2_/*K_a_*_1_)/(1/*T*_1_ − 1/*T*_2_)] × R,(6)
Δ*S* =(Δ*H* − Δ*G*)/*T*,(7)
where *K* is the binding constant (*K_a_*_1_ and *K_a_*_2_ are *K_a_* at temperatures of 293 and 303 K, respectively), R is the gas constant (8.314 J (mol·k)), and *T* is the experimental temperature.

### 4.12. Statistical Analysis

The experimental results were expressed as the mean ± standard deviation (SD). All the experimental data were processed by statistical software-SPSS 23.0, and 1-way variance was used to compare multiple groups of means and 2-way repeated measures ANOVA was used to compare repeated observation data at different time points. Pairwise comparisons were analyzed using the LSD test. Significance was defined as *p* value < 0.05.

## 5. Conclusions

In summary, the data presented in this report clearly demonstrate that the administration of anthocyanins from the fruits of *L. ruthenicum* reduces body weight, lowers serum cholesterol levels, and improves liver steatosis in high-fat-diet-fed mice by inhibiting the activity of pancreatic lipase and altering the intestinal microbiota. Therefore, anthocyanins from the fruits of *L. ruthenicum* may be a functional agent for prevention or improvement of obesity. The underlying mechanisms of *L. ruthenicum* in combating obesity are unclear.

## Figures and Tables

**Figure 1 molecules-27-02141-f001:**
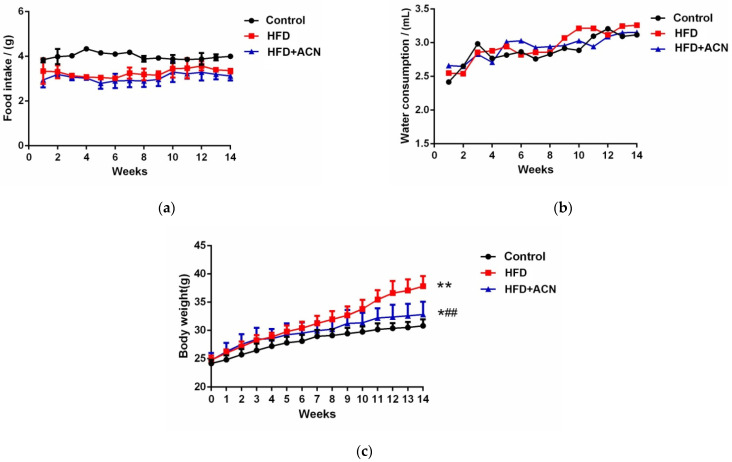
The effects of anthocyanin extract from the fruits of *L. ruthenicum* supplementation on food intake, water consumption, and body weight in high-fat-fed C57BL/6 mice. Data are expressed as the mean ± SD (*n* = 8 for each group), Compared with the control group, * denotes *p* < 0.05; ** denotes *p* < 0.01. Compared with the high-fat diet, ## denotes *p* < 0.01. (**a**) Food intake. (**b**) Water consumption. (**c**) Body weight.

**Figure 2 molecules-27-02141-f002:**
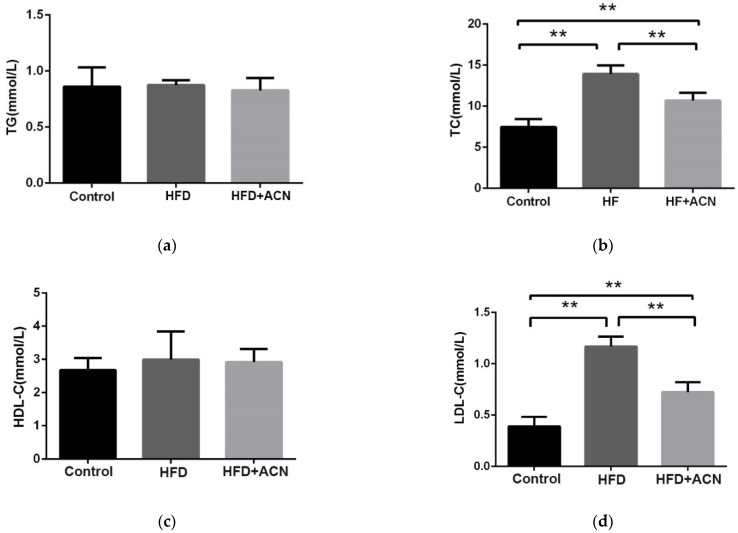
The effects of anthocyanin extract from the fruits of *L. ruthenicum* on the serum lipid levels in high-fat-fed C57BL/6 mice. Data are expressed as the mean ± SD (*n* = 8 for each group), ** denotes *p* < 0.01. (**a**) TG. (**b**) TC. (**c**) HDL-C. (**d**) LDL-C.

**Figure 3 molecules-27-02141-f003:**
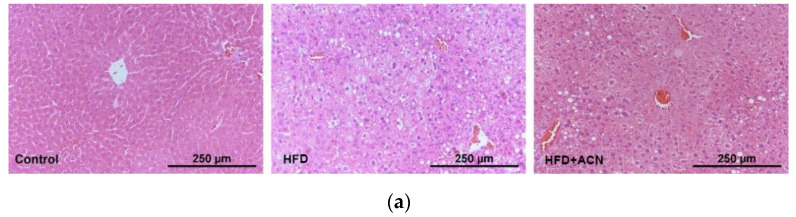
Morphology changes in the liver and adipose tissue. Photo images of H&E and Oil Red O staining of liver sections were photographed at ×200 magnification. Data are expressed as the mean ± SD (*n* = 8 for each group), ** denotes *p* < 0.01. (**a**) H&E staining of liver tissue. (**b**) Oil Red O staining of liver tissue. (**c**) H&E staining of adipose tissue. (**d**) Adipose coefficient.

**Figure 4 molecules-27-02141-f004:**
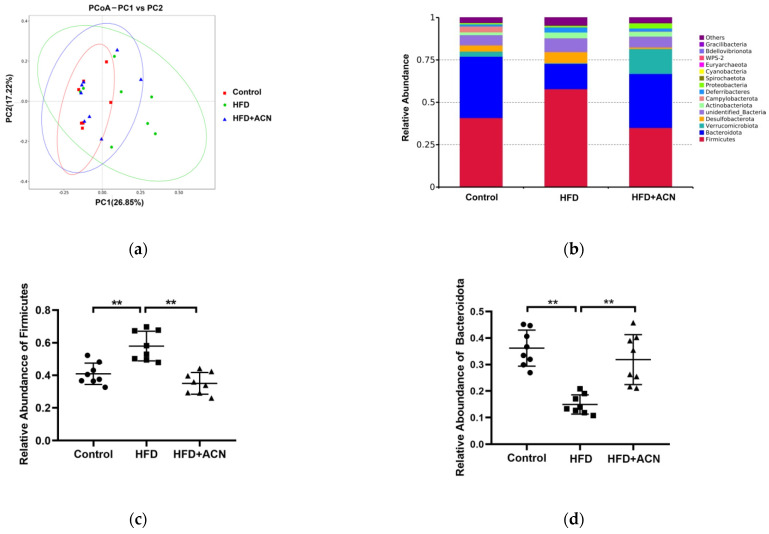
The changes in the gut microbiota structure at the phylum level. Data are expressed as the mean ± SD (*n* = 8 for each group), ** denotes *p* < 0.01. (**a**) PCoA plot. (**b**) Histogram of the relative abundance of the dominant bacterial phyla. (**c**) The relative abundance of *Firmicutes*. (**d**) The relative abundance of *Bacteroidota*. (**e**) The ratio of *Firmicutes* to *Bacteroidota*. (**f**) The relative abundance of *Verrucomicrobiota*.

**Figure 5 molecules-27-02141-f005:**
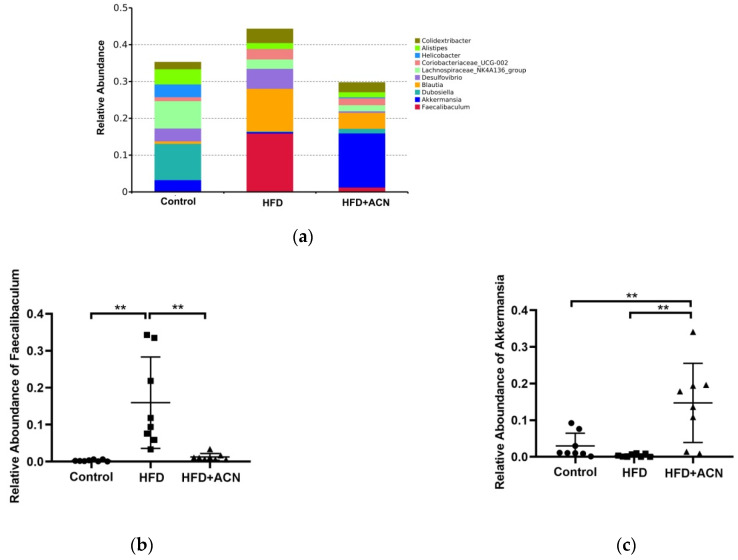
The changes in the gut microbiota structure at the genus level. Data are expressed as the mean ± SD (*n* = 8 for each group), ** denotes *p* < 0.01. (**a**) The accumulative abundance of the top 10 genera among the different groups. (**b**) The relative abundance of *Faecalibaculum*. (**c**) The relative abundance of *Akkermansia*. (**d**) The relative abundance of *Dubosiella*. (**e**) The relative abundance of *Blautia*.

**Figure 6 molecules-27-02141-f006:**
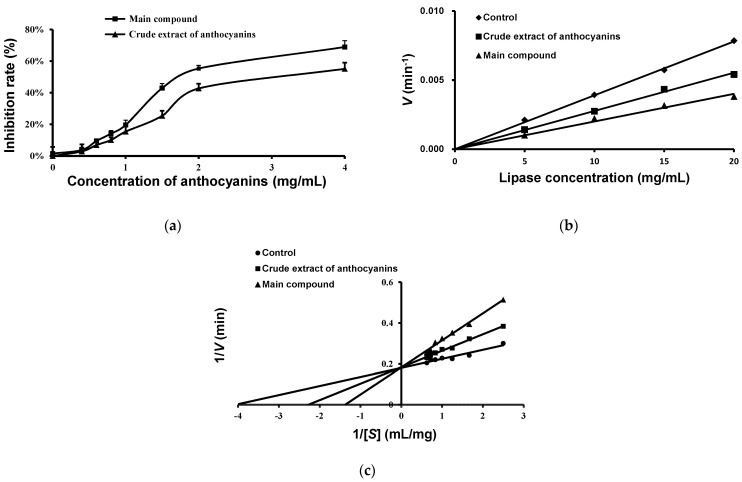
The inhibitory effect of anthocyanins from the fruits of *L*. *ruthenicum* on pancreatic lipase activity. (**a**) The inhibitory effect of anthocyanins on pancreatic lipase activity. (**b**) The inhibitory type of pancreatic lipase by anthocyanins. (**c**) The *Lineweaver–Burk* curve analysis of the inhibition kinetics of the lipase inhibitory effects of anthocyanins.

**Figure 7 molecules-27-02141-f007:**
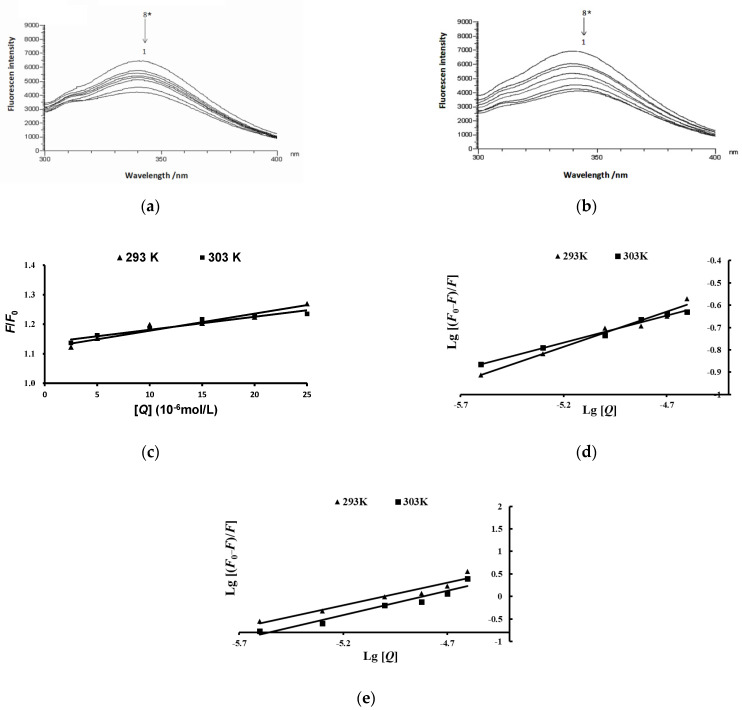
The fluorescence quenching effect of anthocyanins fromthe fruits of *L*. *ruthenicum* on pancreatic lipase. (**a**,**b**) The fluorescence quenching effects of the crude extract of anthocyanins and main compound anthocyanins on pancreatic lipase. * 8→1 The final concentration rates of anthocyanins and pancreatic lipase were 0, 0.5:1, 1:1, 2:1, 4:1, 5:1, 10:1, and 20:1, respectively. (**c**,**d**) The *Stern–Volmer* curve of pancreatic lipase fluorescence quenching by anthocyanins. (**e**) Plot of Lg [(*F*_0_−*F*)/*F*] vs. Lg [*Q*].

**Table 1 molecules-27-02141-t001:** Quality control of data. Data were expressed as the mean ± SD, *n* = 8.

Groups	Raw PE	Effective Tags	AvgLen (nt)	Effective%
Control	102,275.0 ± 6293.5	63,610.9 ± 3458.7	416.6 ± 1.1	62.2 ± 1.7
HFD	103,448.4 ± 6110.5	63,364.3 ± 3605.0	415.4 ± 2.2	61.3 ± 1.4
HFD + ACN	99,454.8 ± 8725.6	61,244.4 ± 4140.6	415.4 ± 1.3	61.7 ± 1.7

**Table 2 molecules-27-02141-t002:** The indexes of α-diversity. Data were expressed as the mean ± SD, *n* = 8.

Groups	Shannon	Simpson	Chao1	ACE
Control	6.6 ± 0.4	1.0 ± 0.0	661.0 ± 87.0	664.4 ± 90.1
HFD	5.9 ± 0.6 *	0.9 ± 0.0	548.9 ± 114.2 *	553.8 ± 116.7
HFD + ACN	6.1 ± 0.4 *	0.9 ± 0.0	637.8 ± 98.6	646.3 ± 99.9

* Compared to control group, denotes *p* < 0.05.

**Table 3 molecules-27-02141-t003:** The IC_50_ and kinetic parameters of pancreatic lipase in the presence of different anthocyanins from the fruit of *L. ruthenicum*.

Groups	IC_50_ (mg/mL)	*V*_max_/min^−1^	*K_m_* (mg/mL)	*K* (mg/mL)
Control	---	5.46	0.25	----
Crude extract of anthocyanins	3.03 ± 0.43	5.46	0.44	3.80
Main compound	1.80 ± 0.20	5.46	0.73	0.88

**Table 4 molecules-27-02141-t004:** The *K_sv_, K_q_, K_a_*, and *n* values of anthocyanins from the fruit of *L. ruthenicum* at different temperatures.

Groups	*T* (K)	*K_sv_* (10^4^ L/mol)	*K_q_* (10^12^ L/(mol·s))	*K_a_* (L/mol)	*n*
Crude extract of anthocyanins	293	0.58	0.58	7.14	0.34
	303	0.44	0.44	3.15	0.24
Main compound	293	6.03	6.03	15.53 × 10^4^	1.08
	303	4.19	4.19	9.69 × 10^4^	0.99

**Table 5 molecules-27-02141-t005:** Thermodynamic analysis of anthocyanins from the fruit of *L. ruthenicum* on pancreatic lipase at different temperatures.

Groups	*T* (K)	Δ*H* (KJ/mol)	Δ*S* (J/mol/K)	Δ*G* (KJ/mol)
Crude extract of anthocyanins	293	−32.57	−12.03	−29.04
	303	−32.57	−12.03	−28.92
Main compound	293	−60.37	−189.70	−4.79
	303	−60.37	−190.01	−2.80

**Table 6 molecules-27-02141-t006:** The composition of the high-fat diet MD12015.

Composition	Proportion
Casein	19.47%
Corn starch	4.99%
Maltodextrin	9.98%
Sucrose	34.11%
Cellulose	4.99%
Corn oil	0.99%
Anhydrous milk fat	19.97%
Cholesterol	0.15%
Small material	5.35%

## Data Availability

All the data that support the findings of this study are available on request from the corresponding author.

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
