# Peer review of "Preventive Effects of Anthocyanins from Lyciumruthenicum Murray in High-Fat Diet-Induced Obese Mice Are Related to the Regulation of Intestinal Microbiota and Inhibition of Pancreatic Lipase Activity"

_molecules, 2022, doi:10.3390/molecules27072141_

Round 1

Reviewer 1 Report

Li et al. studied the effect of Lycium ruthenicum Murray on intestinal microbiota in high fat induced obese mice. Although this work contains novelty; however, this study did not clearly show the anti-obesity effects of Lycium ruthenicum Murray or the underlying mechnisms.

Comments:

The main limitation of this article is that pancreatic lipase alone can not explain the anti-obesity effects of Lycium ruthenicum Murray. The authors only studied pancreatic lipase. Liver and adipose were examined only by histology. Pancreas tissue should be examined microscopically. The role of microbiota looks correlational.

Methods: One important group is missing, i.e., control plus ACN.

Statistics: LSD is not a powerful host hoc examination. Fig 1, 2, 3 should be analyzed by 2-way ANOVA with repeat measures.

Results: The underlying mechanisms of Lycium ruthenicum Murray in combating obesity is unclear.

Some typos: Wu T et al. (line 293) should be Wu et al.,; high fat diet should be abbreviated as HFD;

Author Response

Dear Reviewer,

    Thank you so much for your careful review and constructive suggestions with regard to our manuscript. These comments are all valuable and very helpful for revising and improving our paper, as well as the importance guiding significance to our researches. We have revised our manuscript carefully according to your suggestions. The main corrections and responds to the comments are provided below.

Comment 1. The limitation of this article is that pancreatic lipase alone can not explain the anti-obesity effects of main Lycium ruthenicum Murray. The authors only studied pancreatic lipase. Liver and adipose were examined only by histology. Pancreas tissue should be examined microscopically. The role of microbiota looks correlational.

    Answer: Thank you so much for your suggestions. These comments are all valuable and very helpful for revising and improving our paper, as well as the importance guiding significance to our researches. Due to your suggestion, we have found some shortcomings in our current work. We will improve our research method and achieve more results according to your suggestions in future work. We have added some discussions on this issue in our revised manuscript.

Comment 2. Methods: One important group is missing, i.e., control plus ACN.

    Answer: Thank you so much for your constructive suggestions. We are very sorry for ignoring the control plus ACN. We will add this group in our future work.

Comment 3. Statistics: LSD is not a powerful host hoc examination. Figs 1, 2, 3 should be analyzed using a 2-way ANOVA with repeated measures.

    Answer: Thank you very much for your careful review and constructive suggestions with regard to our manuscript. According to your suggestion, we have made correction in the Figure 1, Figure 2 and Figure 3 by using 2-way ANOVA with repeated measures. We also rewrite this part according to your suggestion.

Comment 4. Results: The underlying mechanisms of Lycium ruthenicum Murray in combating obesity are unclear.

    Answer: Thank you very much for your constructive suggestions with regard to our manuscript. According to your suggestion, we have added this sentence in the Conclusions part.

Comment 5. Some typos: Wu T et al. (line 293) should be Wu et al., the high-fat diet should be abbreviated as HFD;

    Answer: Thank you very much for your constructive suggestions with regard to our manuscript. All of errors like “Wu T et al.” have been corrected. The high-fat diet has been abbreviated as HFD in our manuscript.

The Introduction part also has been improved, please check that in the revised manuscript.

Thank you again for your careful review and constructive suggestions.

Reviewer 2 Report

Na Li et al. demonstrated that Preventive Effects of Anthocyanins from Lycium ruthenicum Murray in High-Fat Diet-Induced Obese Mice are Related to the Regulation of Intestinal Microbiota and Inhibition of Pancreatic Lipase Activity.

Overall the observations are very interesting, the manuscript will be enhanced by describing a point as below. 

  1. The authors state that the weight-loss effect of anthocyanins extracted from the fruit of L. ruthenicum is due to changes in the intestinal microflora. Since anthocyanins extracted from the fruits of L. ruthenicum inhibit pancreatic lipase, it is speculated that fat absorption from the gastrointestinal tract was suppressed, resulting in weight loss and a decrease in blood cholesterol levels. Did you investigate any changes in fecal cholesterol and bile acids?

Author Response

Dear Reviewer,

   Thank you so much for your careful review and constructive suggestions with regard to our manuscript. These comments are all valuable and very helpful for revising and improving our paper, as well as the importance guiding significance to our researches. We have revised our manuscript carefully according to your suggestions. The main corrections and responds to the comments are provided below.

Comment 1. The authors state that the weight-loss effect of anthocyanins extracted from the fruit of L. ruthenicum is due to changes in the intestinal microflora. Since anthocyanins extracted from the fruits of L. ruthenicum inhibit pancreatic lipase, it is speculated that fat absorption from the gastrointestinal tract was suppressed, resulting in weight loss and a decrease in blood cholesterol levels. Did you investigate any changes in fecal cholesterol and bile acids?

    Answer: Thank you very much for your careful review and constructive suggestions with regard to our manuscript. These comments are all valuable and very helpful for revising and improving our paper, as well as the importance guiding significance to our researches. According to your suggestion, we will investigate the effect of anthocyanins extracted from the fruit of L. ruthenicum on the changes in fecal cholesterol and bile acids in our future work.

Reviewer 3 Report

For me this article is complete and scientifically in depth. The experimental protocol, expecially for in vivo treatment, is explained in detail .

Author Response

Dear Reviewer,

      Thank you so much for your careful review and constructive suggestions with regard to our manuscript. These comments are all valuable and very helpful for revising and improving our paper, as well as the importance guiding significance to our researches. We have revised our manuscript carefully according to your suggestions.

Comment 1. For me this article is complete and scientifically in depth. The experimental protocol, expecially for in vivo treatment, is explained in detail.

    Answer: Thank you very much for your suggestions with regard to our manuscript.

Round 2

Reviewer 1 Report

The authors have addressed all my concerns.